# Combatting Climate Change within the EU Green Deal in Contemporary Forestry Administrative Systems: A Case Study of the Umbria Region

**Francesco Barbarese** [1,*] , **Zacharoula S. Andreopoulou** [2] , **Walter Mattioli** [3] , **Loredana Oreti** [4]
**and Francesco Carbone** [1]

1   Department for Innovation in Biological, University of Tuscia, Agro-Food and Forest Systems (DIBAF), 01100 Viterbo, Italy; fcarbone@unitus.it
2   Laboratory of Forest Informatics, Faculty of Forestry and Natural Environment, Aristotle University of Thessaloniki, 54124 Thessaloniki, Greece; randreop@for.auth.gr
3   Council for Agricultural Research and Economics (CREA), Research Centre for Forestry and Wood, Via Valle Della Quistione 27, 00166 Rome, Italy; walter.mattioli@crea.gov.it
4   Council for Agricultural Research and Economics (CREA), Research Centre for Engineering and Agro-Food Processing (CREA-IT), Via Della Pascolare 16, 00015 Monterotondo, Italy; loredana.oreti@crea.gov.it
*   Correspondence: francesco.barbarese@unitus.it

**Abstract:** The integration of digital technologies into forest management is crucial for the European Union's Green Deal, the Forestry Strategy 2030, and Italy's national forestry strategy, aiming to enhance governance and efficiency. However, sustainable forest management's administrative aspects are often overlooked, despite the potential for digital tools to significantly improve environmental performance. Through a Life Cycle Assessment (LCA) for a case study in the Umbria region, the research quantifies $CO_2$ emissions associated with over-threshold forestry administrative procedures under both current and future digitalisation scenarios. Data were collected from legislative documents, interviews with forestry professionals, and emissions calculated according to ISO standards. The findings reveal a considerable reduction in $CO_2$ emissions through digitalisation, from 75.07 $kgCO_2$ to 38.14 $kgCO_2$, with the implementation of the LIFE FOLIAGE project's digital platform. This underscores digitalisation's role in climate change mitigation, highlighting the significant, albeit modest per procedure, of the cumulative national impact. The main findings support further digitalisation in forest management as a method of increasing resource efficiency and reducing greenhouse gas emissions, thus contributing to the Green Deal's and Italy's forest sustainability goals.

**Keywords:** digitalisation; information and communication technologies (ICTs); digital forestry; life cycle assessment (LCA); carbon footprint; forestry administrative procedures; digital platform

## 1. Introduction

### 1.1. Background

In order to implement the strategies outlined in the Green Deal, the European Union (EU) has recently adopted several strategic documents such as the conservation of biodiversity and the reduction in greenhouse gas emissions (GHG) [1] and the EU Forest Strategy 2030 [2]. In particular, the Forestry Strategy [3] stresses the need to monitor the state of forest ecosystems, to provide the framework for the exploitation of forest resources, in order to understand the dynamics of forest areas. From this perspective, the current path of digitalisation, which also involves the forest system, as well as the introduction of information and communication technologies (ICTs), promotes the creation of a shared, coordinated, and inclusive European forest system [4,5]. In order to proceed with the implementation of FS2030 at a national level and following the recent Italian Law on Forests and Forestry Supply Chains [6], Italy has recently published the national forestry strategy [7].

Moreover, in 2018, the National Parliament adopted the "Text on Forests and Forest Chains" in which, among other things, digitalisation was recognised as one of the most important and innovative tools to achieve effective forest governance.

Remote sensing, Geomatics, and Information and Communication Technologies (ICTs) create the conditions for defining a new approach of forest management, known as "precision silviculture" or "precision forestry" [4]. The integration of well-established technologies like remote sensing [8–10] and geographic information systems [11,12] with emerging advancements, such as remotely piloted aerial systems, web-GIS [13], smartphone apps [14], augmented reality, and virtual reality [15], are an opportunity to establish a comprehensive and inclusive forest organisation process. New digital tools have been introduced into management activities, which enable the accurate monitoring, planning, and management of forest resources. These technologies bring about significant positive effects on production quality, cost reduction, and the mitigation of environmental and social impacts [16]. These advanced technologies have a high potential to revolutionise, improve, and make the various processes in the forest system more efficient. This is also due to the fact that important data on the forestry sector is not available for all of Italy [17], particularly when it comes to the quantity of administrative procedures for silvicultural interventions.

The implementation of FS2030 and the drive toward digitalisation have had a tangible effect in Italy, as evidenced by the financing of the "National Forest Information System" (SINFOR) [18] by the Ministry of Agriculture, Food Sovereignty and Forestry (MASAF), an online portal, with free access, in which private individuals, administrations, and public institutions can acquire all the forest information in Italy such as forest area, forest heritage, use of forests, ownership, datasets, etc. [19].

The LIFE FOLIAGE project [20] is set within this context of digitalisation. Its most relevant result is to develop a digital platform named "Forest Management Platform" (hereafter PAF), where public or private forest owners, or their forest consultants, submit technical information about forest management and timber production. There are increasing requests for data and information relating to the state of woody and non-woody forest production and, by successive aggregation, these should be available for forest governance at various institutional levels. The growing climate change tension has led to the promotion of large-scale digitalisation and many interconnected environmental issues, including forest systems. The digitalisation of data and information, etc., ultimately allows forest and environmental governance to be improved [21].

*1.2. State-of-the-Art in Italy*

Considering the previous background, it is important to focus on administrative procedures in the forestry sector and examine how their digitisation might aid in climate change mitigation.

In Italy, forest management is a controlled activity. Even before the unification of Italy (1861), local legislators recognised the importance of forests in the prevention of hydrogeological instability. The recognition of the existence of a significant general interest in forests determines the following:

(a) the definition of the discipline of forest management, through the forestry regulation;
(b) the introduction of administrative procedures to safeguard the general interest in forests.

In the decade from 1990 to 2000, all regions defined their regional discipline of forest management (named Regional Forestry Regulation) and administrative proceedings. Generally, forest owners, users, and logging companies must conform to regional forestry regulations that provide criteria for the determination of forest utilisation projects. Forest administrative procedures vary from region to region, varying in forest management standards, project types, and content. At a national scale, four categories of instances can be identified (Table 1), which are as follows:

(a) sub-threshold instances, the forest management of which covers small areas;

(b)  over-threshold instances, the management of which covers areas larger than those below the threshold, but not exceeding the area indicated in the regional forestry regulation;

(c)  instances in derogation, which concern management interventions in areas with significant forest and environmental characteristics;

(d)  instances of implementation of forest management plans, which relate to the management of forest plots according to the specific standard set out in the plan.

**Table 1.** Type of forestry instances.

| Instances | Forest Area Under Management | Forest Management Standards | Project Types | Forest Institutions Involved | Administrative Procedure | Type of Permission |
|---|---|---|---|---|---|---|
| Sub-threshold | Small area | Regional Forest Regulation | No project | Forest office | Simple | Silent |
| Over-threshold | Medium area | Regional Forest Regulation | Project | Forest Office | Accuracy | Silent or Overt |
| Instances in derogation | High forest and environmental value area | Regional Forest Regulation integrated with other disciplines | Complex Projects | Forest Office and other institution | Complex and inter-institutional | Overt |
| Instances of implementation of forest management plans | Forest plots | Forest Management plan | Project details | Forest Office | Simple | Silent |

Several authors recognise that digitalisation contributes to the fight against climate change, providing services that, in the past, were provided with significant GHG emissions [16,21–27]. This is also the expectation for the Italian forest system, even if Carbone et al. registered that only 8 of the 21 Autonomous Regions and Provinces had an advanced degree and level of digitalisation, in the year 2020 [28].

Forestry administrative procedures involve many actors, forest owners, forestry consultants, forestry offices, and forestry services. They use tools and instruments and develop activities that produce GHG emissions.

As a result, we attempted a case study of an above-threshold forestry administrative operation and calculated its $CO_2$ emissions by comparing two different scenarios. The first scenario evaluates the state of digitalisation in 2020, while the second forecasts the prospects for 2025. By this date, the dematerialisation goals of administrative procedures will have been achieved and the forest administrative procedures will have minimised their impact on the capacity of forests to absorb $CO_2$. Using the LCA method, the study estimates the $CO_2$ emitted of an over-threshold administrative procedure of the Umbria Region.

## 2. Methodology

The carbon footprint is an indicator for measuring the total $CO_2$ emissions generated by human activity to obtain goods and services and to develop activities. Life Cycle Assessment (LCA) is an important and fundamental methodology for quantifying the environmental impacts arising from the production of goods or the provision of services. It is largely used as tool for combatting climate change [29]. This study combines the carbon footprint and LCA methodologies to quantify the $CO_2$ emissions related to one administrative procedure for the management of a forest in the Umbria region. Data were gathered between 2022 and 2023 from regional legislative documents, including administrative procedures for forest management, as well as their relative forms and involved actors, through questionnaires and interviews with forestry professionals and

relevant offices. The emissions calculation method is defined in accordance with the standards ISO 14067 [30]. The methodology used is shown in Figure 1.

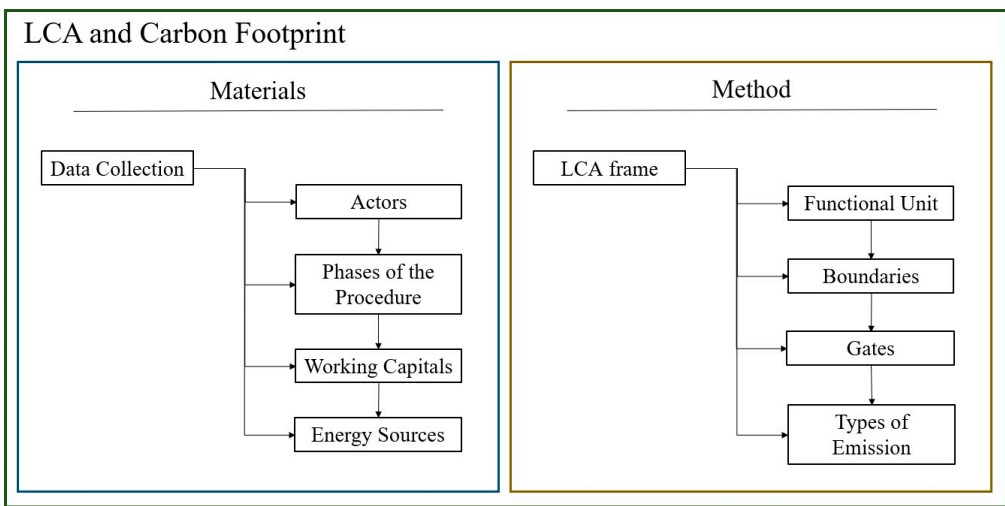

**Figure 1.** Methodology for calculating the carbon footprint of a forest administrative operation using the LCA process.

*2.1. Materials*

2.1.1. Data Collection

In the Umbria region, forest management is regulated by the legislative decree 34/2018 [6], the regional law 28/2001 [31], and the regional forest regulation 7/2002 [32]. The main information collected from regional legislative texts is as follows:

- administrative procedures for obtaining the permission to implement forest management;
- forms required to fulfil the administrative procedure;
- actors involved in the various stages of the procedure.

All the primary data collected from legislative documents, through questionnaires and interviews conducted with forestry professionals and the competent offices of the forestry sector, are as follows:

- tools and the instruments used throughout the 'lifecycle' of the procedure;
- hours the tools were used;
- average time taken for each step in the implementation of the procedure.

The data units and sources for each process flow were carefully selected and evaluated.

2.1.2. Actors

There are four types of actors involved in the development of the administrative procedure, as follows:

- Forestry Professionals (FPs)—qualified experts dedicated to promoting responsible land and resource management practices, fostering sustainable ecosystems, and ensuring the long-term health of forests and natural environments. FPs are those who make the most use of instruments and tools, such as computers, measuring instruments, and also vehicles, to reach and explore forest areas. The main activities to be carried out are the drafting of the project, on-site inspections, and monitoring of the work.
- Regional Forestry Agency (AFOR)—a public agency, established and controlled by the Umbria Region. It carries out functions assigned to it by Regional Law and subsequent amendments and additions. Specifically, its responsibilities include managing agro-forestry assets belonging to the region, activities for the protection and enhancement of existing forests, and measures for the prevention and control of fires. The AFOR takes care of the formal analysis of the forest documentation submitted and the issuing of the project approval documentation at the end of administrative procedures. In

carrying out these actions, it makes use of vehicles for the various journeys, as well as office equipment.

- Forestry, Environment and Agri-Food Unit (CUFAA)— they carry out activities in the field of environmental, territorial, and water protection, as well as in the agro-food and forestry sectors. Their main activities are the monitoring and control of the state of the natural capital of the forest from the start to the end of its management. The CUFAA makes large use of vehicles, draws up reports, and records data related to the forestry system.
- Logging Companies (LCs)—the entities that manage the forest using dedicated instruments (chain saws, tractors, trucks, and other minor tools). They are registered on a specific regional list. For their operations, LCs use vehicles and other typical office tools.

### 2.1.3. Administrative Forest Procedures

Three main phases have been delineated, as shown in Figure 2 and as follows:

- acquiring permits to manage the forest;
- operational phase, involving only the administrative forest procedures and net $CO_2$ emissions emitted from the use of dedicated forest instruments (chains saws, tractors, trucks, etc.);
- monitoring the area subject to forest use.

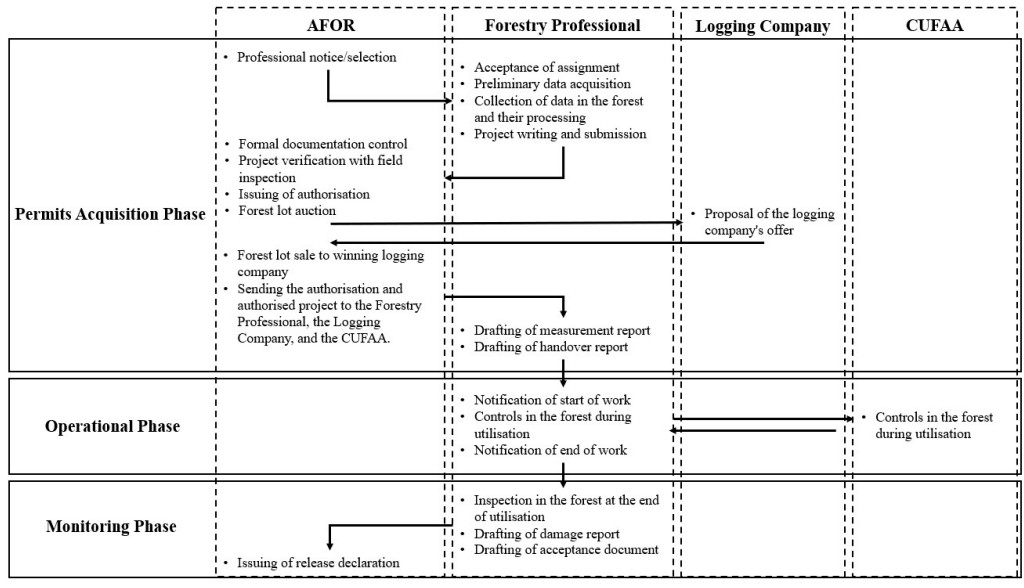

**Figure 2.** Process flow of the Forestry Administrative Procedures in Umbria.

For the management of a forest area of more than 5 hectares, it is mandatory to submit a project, drawn up by a forestry professional, that complies with the provisions of the regional forestry regulation. Once the adequacy of the project has been verified by the AFOR Forest Office, the latter can give its approval (overt permission) or, after 90 days from the submission, this is tacitly approved (silent permission). At the same time, the forestry office (AFOR) holds an auction to allocate the forest lot for cutting to a logging company. Once the procedures for assigning the forest lot are completed, the forestry professional draws up the measurement and handover reports, concluding the permits acquisition phase.

In the operational phase, the administrative procedure is very reduced. Under normal conditions, it mainly involves the FP, who must draw up the notification document for the start of the works, carry out the forest controls during the use of the forests, and draw up the notification document for the end of the work.

The last stage of forest administrative procedures is monitoring. Once the forest management is complete, the forestry professional checks whether the management intervention meets expectations. If so, the AFOR issues the declaration of completion of the works and the closure of the administrative procedure.

### 2.1.4. Working Capital

Decarbonisation and dematerialisation are the main targets of the forest administrative procedures. At the same time, the introduction of new high-tech forest tools, new and more articulated forest platforms for the collection and sharing of data, as well as the storage of these in cloud computing, will be the expectations of the near future.

Tools, instruments, and materials make up the working capital used in the administrative forest procedures. Of the 12 instruments mentioned in Table 2, at the beginning of the decade, 8 were fundamental to the administrative procedures and all were included in the categories of instruments and materials. The toolkit, such as the cloud and drones, was emerging as a possible working tool, while computers/laptops evolved from passive writing to dynamic tools that capture information and data for cloud storage. As for vehicles, there was pressure to replace them with those of lower environmental impact, with low or zero $CO_2$ emissions.

**Table 2.** Main tools, instruments, and materials used in forestry administrative procedures.

|  |  | Tools, Instruments, and Materials Used | |
|---|---|---|---|
|  |  | **Currently** | **In the Near Future** |
| Tools | Printer | Y | N |
|  | Dendrometer stand | Y, A | Y, D |
|  | Forest Hypsometer | Y, A | Y, D |
|  | Measuring tape | Y, A | Y, D |
|  | Smartphone | Y | Y |
| Instruments | Laptop | Y | Y |
|  | Internet and cloud computing | N | Y |
|  | Drone | N | Y |
|  | Forest vehicles | Y, F | Y, E |
| Materials | Pen | Y | N |
|  | Paper sheet | Y | N |
|  | Spray Paint Can | Y | N |

Legend: Y = used, N = not used; A = analogue, D = digital; F = fossil fuel, E = environmentally high-performing fuels.

Based on data found in the scientific literature; in technical manuals of tools, instruments, and materials; and from conversation with experts and subsequent elaborations, the unitary emissions of $CO_2$ have been presented in Table 3.

### 2.1.5. Energy Sources

The assessment of $CO_2$ emissions associated with the use of energy sources, including both electricity and fuels, was conducted by referencing the report edited by ISPRA [33], as well as the technical manuals provided by the vehicle manufacturer. These resources provide emission coefficients that are specific to each energy source, thereby facilitating an accurate evaluation of the corresponding $CO_2$ emissions (Table 4).

**Table 3.** $CO_2$ emissions for the usage, construction, and consumption of tools, instruments, and materials.

| | | Emissions | | | | |
|---|---|---|---|---|---|---|
| | | Usage | | Construction | | Consume |
| | | kgCO$_2$/h | kgCO$_2$/h | kgCO$_2$/h | kgCO$_2$/km | kgCO$_2$/n |
| Tools | Printer | 0.00219 | | 0.01096 | | |
| | Dendrometer stand | 0.05500 | | 0.00003 | | |
| | Forest hypsometer | 0.05500 | | 0.00057 | | |
| | Measuring tape | 0.05500 | | 0.00014 | | |
| | Smartphone | 0.05500 | | 0.00716 [C]<br>0.00727 [NF] | | |
| Instruments | Laptop | 0.00232 [C]<br>0.00215 [NF] | | 0.03852 [C]<br>0.02761 [NF] | | |
| | Internet and Cloud computing | 0.02000 | | | | |
| | Drone | 1.38240 | | | | |
| | Forest vehicles | | 0.15900 [C]<br>0.07245 [NF] | | 0.07863 | |
| Material | Pen | | | | | 0.14000 |
| | Paper Sheet | | | | | 0.00464 |
| | Spray Paint Can | | | | | 0.46000 |

Legend: [C] = currently; [NF] = near future. Sources: Technical manuals and conversations with experts.

**Table 4.** $CO_2$ emissions related to energy sources used.

| Type of Energy Sources | Emissions | |
|---|---|---|
| | kgCO$_2$/L | kgCO$_2$/kWh |
| Diesel | 2.65 | |
| PLG | 1.61 | |
| Electricity (2020) | | 0.55 |
| Electricity (2022) | | 0.51 |

Sources: [33]; technical manuals.

*2.2. Methodology*

2.2.1. Functional Unit

The functional unit subjected to LCA analysis is an over-threshold instance relating to a mixed coppice of public ownership, in the Umbria Region. The forest area, a mixed wood of *Quercus ilex* L., *Ostrya carpinifolia* Scop., and *Fraxinus ornus* L., is classified as being a medium area (10 hectares), that is, of extension, advanced to the threshold of 5 hectares (sub-threshold) (Table 1). It is not internal to protected areas and the management that will be implemented is consistent with what is indicated by the Regional Forest Regulations. Administratively, the property must submit a forest management project drawn up by a forestry professional. The competent institution is the AFOR, which will have to proceed accurately according to procedure. After that, if after 90 days the owner does not receive the formal decision from the AFOR (silent permission), they can begin the work. This is the typical project submitted by forest owners.

2.2.2. Boundaries

The starting point of the LCA is the decision of the forest owner to proceed with the management activity. For this purpose it is assumed that the forest is publicly owned

and located in the Umbria Region. The end of the administrative procedure is identifiable in the delivery of the monitoring certification issued by the forest office. This document declares that forest management has been carried out in a correct and consistent manner, with the management standard defined from Forest Law, regulations, and the forest management project.

### 2.2.3. Gates

The LCA is developed using the "multiple gates" approach that identifies three intermediate gates, which are (a) permit acquisition phase; (b) operational phase; and (c) monitoring phase. These three gates define the overall process of the administrative forest system.

The starting point of this is the decision of the forest owner to perform forest management, while the term is identified in the verification of the management monitoring. The aggregation of the $CO_2$ emissions of the three gates provides the overall emissions of the administrative forest procedure.

### 2.2.4. Types of Emissions

$CO_2$ emissions fall into the following categories:

- direct emissions—those due to the direct use of working capital;
- collateral emissions—those due to the use of working capital as a means to achieve the objective;
- remote emissions—those issued when working capital equipment was built for direct or collateral use.

### *2.3. Elaboration*

Direct emissions (Equation (1)) are due to the use of tools and instruments in the administrative procedures. They depend on the type of energy source and the time allocated for the execution of the individual procedure.

$$DE = E_{\frac{kgCO_2}{h}} \times T_h \tag{1}$$

where

- $DE$: direct emissions
- $E_{kgCO_2/h}$: use emissions
- $T_h$: use time

In the category of collateral emissions, an important source is paper and vehicles. Currently, projects, permissions, communications, exchange information, etc., require a large amount of paper. The emissions of $CO_2$ for the production of documents has been reported by Dias and Arroja [34]. However, the number of printouts differ greatly according to actors. Additionally, the emissions are calculated based on the use of the vehicle during the carrying out of administrative procedures (Equations (2) and (3)). The sum of the results gives the value of the collateral emissions (Equation (4)).

$$VU = E_{\text{kgCO}_2/\text{km}} \times D_{km} \tag{2}$$

where

- $VU$: vehicle use emissions
- $E_{\text{kgCO}_2/\text{km}}$: vehicle use emissions
- $D_{km}$: distance travelled

$$RCE = \sum UE \times n. \tag{3}$$

where

- *RCE*: rapid consumption tools and materials emissions
- *UE*: unitary emissions of the rapid consumption tools and materials
- *n*.: number of tools and/or materials

$$CE = \sum VU + RCE \tag{4}$$

where

- *CE*: collateral emissions
- *VU*: vehicle use emissions
- *RCE*: rapid consumption tools and materials emissions

Remote emissions are those generated in the process of constructing tools and instruments. In general, considering the emissions generated in the production process, taking into account their working lifespan or the obsolescence time due to intervening innovations, and also considering their average annual usage in workdays, distributed by daily working hours, the magnitude of remote emissions for tools or instruments is quantified. Assuming the total emissions for the production of instruments by individual actors (*RE*), given the working lifespan, the average annual usage in hours, as well as the hours typically dedicated to each forestry administrative procedure or the distance travelled (*U*), the value of the remote emissions have been estimated. Formally, it is presented as follows in Equation (5):

$$RE = DE \times U \tag{5}$$

where

- *RE*: remote emissions
- *DE*: depreciation charge emissions
- *U*: utilisation factor (h or km)

### 3. Results

The direct, collateral, and remote emissions have been categorised based on the digitalisation scenario into those current (intermediate digitalisation) and those of the near future (advanced digitalisation). Table 5 reports the emission values generated by the actors involved in the forest administrative procedures. Currently, the highest emissions have been generated from forest professionals, with 33.50 $kgCO_2$ emitted; it is expected that, once full digitalisation is adopted, it will be 21.55 $kgCO_2$. Then, the AFOR reports 21.44 $kgCO_2$ emitted currently and 11.76 $kgCO_2$ is expected for the near future. The logging company shows the greatest reduction in emissions, from approximately 15 $kgCO_2$ emitted currently to almost no emissions in the near future. On the other hand, the CUFAA see a slight reduction in their emissions, from 5.58 to 4.78 $kgCO_2$.

**Table 5.** Total emissions from the actors involved in the forestry administrative procedure.

| Actors | Direct Emissions | | Collateral Emissions | | Remote Emissions | | Total Emissions | |
|---|---|---|---|---|---|---|---|---|
| | Currently | Near Future | Currently | Near Future | Currently | Near Future | Currently | Near Future |
| | $kgCO_2$ | | $kgCO_2$ | | $kgCO_2$ | | $kgCO_2$ | |
| Forestry Professional | 2.39 | 7.30 | 20.51 | 6.52 | 10.60 | 7.73 | 33.50 | 21.55 |
| AFOR | 1.48 | 4.15 | 13.07 | 3.48 | 6.89 | 4.13 | 21.44 | 11.76 |
| CUFAA | 0.62 | 1.70 | 3.32 | 1.45 | 1.63 | 1.63 | 5.58 | 4.78 |
| Logging Company | 0.02 | 0.02 | 9.70 | 0 | 4.76 | 0.03 | 14.48 | 0.05 |

Overall, throughout the entire administrative process, from the decision to manage the forest to the completion of the administrative acts of the correct execution of the

intervention, the amount of GHGs emitted is 75 kgCO$_2$. When a more efficient level of digitisation is achieved, thanks to the PAF platform, the emissions should be halved (38.14 kgCO$_2$).

Concerning the emissions from dedicated forest tools, instruments, and materials (Table 6), the highest emissive sources are the instruments with 69.03 kgCO$_2$ emitted currently and 32.20 kgCO$_2$ estimated for the near future. Tools, instead, registered an increase in emissions in the near future, from the current 4.94 to 5.94 kgCO$_2$, due to direct emissions produced from the implementation and use of digitalised field tools, primarily from the drone. Finally, the CO$_2$ emissions generated by materials decrease from 1.11 kgCO$_2$, in the current scenario, to zero emissions, explainable mainly by the dematerialisation of paper documents and the consequent redundancy of pens.

**Table 6.** Total emissions produced from tools to support administrative procedures.

| | | Direct Emissions | | Collateral Emissions | | Remote Emissions | | Total Emissions | |
|---|---|---|---|---|---|---|---|---|---|
| | | Currently | Near Future | Currently | Near Future | Currently | Near Future | Currently | Near Future |
| | | kgCO$_2$ | | kgCO$_2$ | | kgCO$_2$ | | kgCO$_2$ | |
| Sources | Tools | 3.60 | 5.64 | 0.98 | 0.00 | 0.36 | 0.30 | 4.94 | 5.94 |
| | Instruments | 0.92 | 7.54 | 44.52 | 11.45 | 23.59 | 13.21 | 69.03 | 32.20 |
| | Materials | 0.00 | 0.00 | 1.11 | 0.00 | 0.00 | 0.00 | 1.11 | 0.00 |

By performing the analysis across the administrative procedure stages (Table 7), the highest emissions are observed during the first phase that starts from the decision to manage the forest to the acquisition of the permit. The emissions estimated are currently 56.76 kgCO$_2$, which, in perspective, are halved, mainly for the dematerialisation of paper documents. The monitoring system records emissions of just under 10 kgCO$_2$, especially due to the use of the vehicle, which will halve for switching to low-emission vehicles, while the operational phase of execution of the works generates rather small emissions, due to the control and supervision of CUFAA activities.

**Table 7.** Total CO$_2$ emissions divided according to the phases in which the administrative procedure is organised.

| | | Direct Emissions | | Collateral Emissions | | Remote Emissions | | Total Emissions | |
|---|---|---|---|---|---|---|---|---|---|
| | | kgCO$_2$ | | kgCO$_2$ | | kgCO$_2$ | | kgCO$_2$ | |
| | | Currently | Near Future | Currently | Near Future | Currently | Near Future | Currently | Near Future |
| Phases of the Administrative Procedure | Permits Acquisition Phase | 3.23 | 11.12 | 35.20 | 7.82 | 18.34 | 9.43 | 56.76 | 28.37 |
| | Operational Phase | 0.64 | 1.73 | 5.38 | 1.45 | 2.58 | 1.66 | 8.60 | 4.83 |
| | Monitoring Phase | 0.64 | 0.33 | 6.02 | 2.17 | 2.96 | 2.43 | 9.63 | 4.94 |

## 4. Discussion

The current organisation of forest administrative procedures has an important digital gap. The main emissive sources are the instruments. This item includes, in particular, vehicles for field measurements, on-site inspections relating to the administrative procedures, daily trips to the work area, and inspections necessary to verify the correct execution of forest management. Finally, the use of vehicles largely explains the emissions of 69 kgCO$_2$.

The other tools and materials result in low emissions of 6 kgCO$_2$. The total emissions are 75.07 kgCO$_2$ (Table 8). Analysis of the actions shows that the drafting of the project is the action that emits the largest volume of CO$_2$, at 35.50 kgCO$_2$. The other actions generate progressively lower emissions, 24.50 and 15.07 kgCO$_2$, respectively, for administrative activities and the verification of the intervention.

**Table 8.** Flow of CO$_2$ emissions from the current digitalisation scenario, based on sources and actions.

| | | Actions | | | |
| --- | --- | --- | --- | --- | --- |
| | | **Forest Management Project** | **Administration Activities** | **Inspection of Forest Area Managed** | **Total Sources** |
| **Sources** | Tools | 1.61 | 1.74 | 1.59 | 4.94 |
| | Instruments | 32.89 | 22.68 | 13.46 | 69.03 |
| | Materials | 1.00 | 0.08 | 0.02 | 1.11 |
| | Total actions | 35.50 | 24.50 | 15.07 | 75.07 |

The near future scenario, with the introduction of the PAF platform, shows that emissions should be halved overall (Table 9). The instruments are the most emissive sources (32.20 kgCO$_2$) and the drafting of the management project is the action that generates the most emissions (16.64 kgCO$_2$). Interestingly, an amount of zero emissions for the materials is highlighted. Overall, the total emissions are 38.14 kgCO$_2$, recording a reduction of 97% compared to the current scenario.

**Table 9.** Flow of CO$_2$ emissions from the near future digitalisation scenario based on sources and actions.

| | | Actions | | | |
| --- | --- | --- | --- | --- | --- |
| | | **Forest Management Project** | **Administration Activities** | **Inspection of Forest Area Managed** | **Total Sources** |
| **Sources** | Tools | 3.97 | 1.32 | 0.66 | 5.94 |
| | Instruments | 12.68 | 10.49 | 9.04 | 32.20 |
| | Materials | 0.00 | 0.00 | 0.00 | 0.00 |
| | Total actions | 16.64 | 11.81 | 9.69 | 38.14 |

Overall, emissions for a forest administrative procedure are reduced from 75.07 kgCO$_2$ in the current digitalisation scenario to 38.14 kgCO$_2$ in the near future digitalisation scenario, for a total reduction of 36.85 kgCO$_2$.

This study was based on the administrative procedure for a forest over-threshold, which was subject to forest management of final rotation. The AFOR estimates that about 150 projects annually are submitted to the agency. As a result, the total emissions released into the atmosphere currently amount to about 11,260 kgCO$_2$, while, in the future, it is expected that, following the introduction of the PAF platform, these should be reduced to 5721 kgCO$_2$ (Table 10).

The case study highlights that the digitalisation of administrative procedures within the forest system can contribute to mitigating climate change, albeit in terms of avoided emissions. Overall, the subject matter has received little attention, likely due to the relatively small numbers associated with individual administrative procedures. However, these become significant when aggregated on a larger territorial scale. This is evident, when considering the Italian experience. Since 1923, the legislature has recognised the public interest in forests, introducing the obligation of administrative procedures as the basis for

forest management. In light of the forest management interventions during 2017–2018, regardless of the types of procedures, it is estimated that around 38,000 administrative procedures were initiated (Our data analysis is based on the Report on the State of Forests and the Forest Sector in Italy, 2017–2018.). Assuming that each procedure currently generates emissions of 75.07 $kgCO_2$ on a national scale, approximately 2850 $tCO_2$ are released into the atmosphere. With the perspective of a national-scale implementation of the PAF platform, this could potentially be reduced to about 1406 $tCO_2$, resulting in avoided emissions of 1444 $tCO_2$.

**Table 10.** Flow of $CO_2$ emissions from the near future digitalisation scenario for the annually submitted projects, based on sources and actions.

|  |  | Total Emissions | |
|---|---|---|---|
|  |  | No. 1 | No. 150 |
| Current Scenario | $kgCO_2$ | 75.07 | 11,260.81 |
| Near future Scenario | $kgCO_2$ | 38.14 | 5721.41 |
| Contribution to Combatting Climate Change | $kgCO_2$ | −36.93 | −5539.40 |

The avoided emissions could further increase with initiatives not directly linked to FOLIAGE. The most significant measure would involve the use of energy derived from renewable sources, especially for the operation of tools, primarily computers, various devices, and vehicles [26,35]. However, the estimated volume of emissions should be considered indicative. This primarily stems from the scale of the study, which focuses solely on the administrative procedures in the Umbria Region, known for having a rather simple and linear forest administrative structure. However, each region has a different forest administrative organisation, that sometimes also involve other territorial institutions, such as provinces and municipalities. Additionally, the considered instance, although among the most common, does not capture the specificities of other instances. It should be noted that the forest administrative procedures are proportionate to the size of the management intervention and environmental values, as evident from Table 1. In particular, instances concerning forests in specific environmental contexts require additional reports beyond mere forest planning. Lastly, it is important to highlight that there are still some reluctances in providing the necessary data for the development of LCA, especially when referring to studies on a smaller territorial scale.

## 5. Conclusions

The development of digitalised systems is a widely supported strategy, particularly in areas where it can lead to more efficient resource use and lower greenhouse gas emissions. This research falls under this category of analysis. Through this case study, it was estimated what the contribution, in terms of reducing greenhouse gases, could be. The advancement in the level of digitalisation of the Umbria Region through the PAF platform would contribute to climate change mitigation to the order of 5500 $kgCO_2$ in avoided emissions per year.

The study of administrative procedures is explained by the sheer number of administrative procedures activated on a national scale, rather than the quantity of emissions characterising each individual process. The total amount of annual avoided emissions should be in the thousands of tonnes. Despite all the limitations for estimating $CO_2$ emissions due to the limited scale of the analysis, which only considers the Umbria region, and the specificity of the administrative instance, without, therefore, considering other types of instances, this study opens numerous research opportunities. To begin, the differentiated impact of digitalisation process across various types of administrative procedures is critical, as are the benefits that plants may potentially experience in terms of their positioning in the carbon credit market.

Furthermore, this type of analysis might be used to evaluate the various procedures utilised in the Umbria region, as well as any forestry administrative procedures in other Italian regions and provinces.

Another important theme would be the social impact generated by digitalisation through the PAF platform, with a focus on the project's partner regions (Umbria and Lazio), a topic to be explored in further studies.

**Author Contributions:** Conceptualisation, F.B.; methodology, F.B., F.C. and W.M.; formal analysis, F.B.; investigation, F.B.; resources, F.C. and Z.S.A.; data curation, F.B.; writing—original draft, F.B.; writing—review and editing, F.C., Z.S.A., L.O. and W.M.; supervision, F.C.; project administration, Z.S.A. All authors have read and agreed to the published version of the manuscript.

**Funding:** This research was funded by the LIFE19 GIE/IT/000311 LIFE FOLIAGE project "Forest planning and earth observation for a well-grounded governance".

**Data Availability Statement:** The data presented in this study are openly available, after direct request to the authors.

**Acknowledgments:** The authors are grateful to the Regional Forestry Agency (AFOR), the Umbria Region, and the Life FOLIAGE project, who provided useful information and technical data and contributed to the success of this study.

**Conflicts of Interest:** The authors declare no conflicts of interest.

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
