# Peer review of "Combatting Climate Change within the EU Green Deal in Contemporary Forestry Administrative Systems: A Case Study of the Umbria Region"

_forests, doi:10.3390/f15050745_

Round 1
Reviewer 1 Report
Comments and Suggestions for Authors
The article should be revised to make it better and really understood easily

Author Response
Thank you very much for taking the time to review this manuscript. Please find the detailed responses below and the corresponding revisions/corrections highlighted/in track changes in the re-submitted files
- Table 1 summarizes the current types of forest instances in Italy for forest management, providing only a general overview.
- I have added a sentence (line 134) specifying that the research was conducted in the Umbria region between 2022 and 2023.
- I have detailed the type of forest, with the main forest species, that was examined in the study (lines 261-262).
- Umbria is a region, one of the 20 administrative territorial entities into which Italy is divided.
- I have included both in the text (lines 231-232) and under Table 3 the sources from which I obtained the data.
- I used the same procedure as for Table 3 for Table 4, inserting the sources of the data in the text and under the table.
- The formulas were developed for calculating CO2 emissions by the authors of the study.
- I have corrected the error of the double Table 9, renumbering the last table as "Table 10."

Reviewer 2 Report
Comments and Suggestions for Authors
Subsection 1.2, State-of-art in Italy contains only a few references. Therefore this subsection should be enriched by adding more references.
The equations should be numbered e.g. (1) etc.
lines 297-298: the sentence lacks the predicate.
line 310: "due to the use of vehicle" is not correct
line 163: "There are four types of actors" instead of "There are four types of actor"
in equation 3: "no" instead of "n"
The conclusions lack the limitation of the study.
line 286: "Types of emissions" instead of "Types of emission"
line 480: "should be in thousands of tonnes." instead of "should be in the thousands of tonnes."
Comments on the Quality of English Language
Subsection 1.2, State-of-art in Italy contains only a few references. Therefore this subsection should be enriched by adding more references.
The equations should be numbered e.g. (1) etc.
lines 297-298: the sentence lacks the predicate.
line 310: "due to the use of vehicle" is not correct.
line 163: "There are four types of actors" instead of "There are four types of actor"
in equation 3: "no" instead of "n"
The conclusions lack the limitation of the study.
line 286: "Types of emissions" instead of "Types of emission"
line 480: "should be in thousands of tonnes." instead of "should be in the thousands of tonnes."
Author Response
Thank you very much for taking the time to review this manuscript. Please find the detailed responses below and the corresponding revisions/corrections highlighted/in track changes in the re-submitted files
- I have added more citations in subsection 1.2 (lines 110-112) to provide a better overview of the state of the art regarding the effects of digitalization on the environment and GHG emissions.
- I have numbered the equations in ascending order following your suggestion.
- I removed the sentence without a predicate and have inserted (Equation no. 1) in line 300.
- I corrected "due to the use of vehicle" to "based on the use of vehicle."
- I corrected the error in the sentence on line 163 (now line 164).
- I corrected the error in Equation 3.
- I added a paragraph in the conclusions regarding the main limitations of the study (Lines 485-488).
- I corrected the error on line 286 (now line 291).
- I corrected the error on line 480 (now line 485).

Round 2
Reviewer 2 Report
Comments and Suggestions for Authors
Using a Life Cycle Assessment (LCA) conducted for a case study in the Umbria region, this research measures CO2 emissions linked to forestry administrative procedures that exceed thresholds under both current and projected digitalization scenarios.
The paper brings a novel approach to forestry administrative procedures.
I think it can be accepted for publication.